# Correlation of Plasmatic Amyloid Beta Peptides (Aβ-40, Aβ-42) with Myocardial Injury and Inflammatory Biomarkers in Acute Coronary Syndrome

**DOI:** 10.3390/jcm13041117

**Published:** 2024-02-16

**Authors:** Luis Eduardo Del Moral, Claudia Lerma, Héctor González-Pacheco, Alan Cristhian Chávez-Lázaro, Felipe Massó, Emma Rodriguez

**Affiliations:** 1Translacional Research Unit, Instituto Nacional de Cardiología Ignacio Chávez, Mexico City 14080, Mexico; luisdelmoral.contact@gmail.com (L.E.D.M.); alancristhianchavezlazaro1@gmail.com (A.C.C.-L.); f_masso@yahoo.com (F.M.); 2Department of Molecular Biology, Instituto Nacional de Cardiología Ignacio Chávez, Mexico City 14080, Mexico; dr.claudialerma@gmail.com; 3Coronary Care Unit, Instituto Nacional de Cardiología Ignacio Chávez, Mexico City 14080, Mexico; hectorglezp@hotmail.com

**Keywords:** acute myocardial infarction, beta amyloid, Aβ-42, Aβ-40, NT-proBNP

## Abstract

**Background/Objective:** Amyloid beta (β) -40 levels increase with age and inflammation states and appear to be associated with clinical manifestations of acute coronary syndrome (ACS). We investigated the correlation of Aβ peptides with myocardial injury and inflammation biomarkers in patients with or without ST elevation myocardial infarction (STEMI, NSTEMI). **Methods:** This singe-center, cross-sectional, observational, and correlation study included 65 patients with ACS (*n* = 34 STEMI, 29 males, age = 58 ± 12 years; *n* = 31 NSTEMI, 22 males, age = 60 ± 12 years) who were enrolled in the coronary care unit within 12 h after symptom onset from February 2022 to May 2023. Aβ peptide levels and biochemical parameters were assessed. **Results:** NSTEMI patients had a higher prevalence of hypertension (*p =* 0.039), diabetes (*p =* 0.043), smoking (*p =* 0.003), and prior myocardial infarction (*p =* 0.010) compared to STEMI patients. We observed a higher level of Aβ-42 in NSTEMI (*p =* 0.001) but no difference in Aβ-40 levels. We also found a correlation between age and NT-proBNP with both Aβ peptides (Aβ-40, Aβ-42) (*p =* 0.001, *p =* 0.002 respectively). **Conclusions:** Our results show that patients with NSTEMI had a higher prevalence of cardiovascular risk factors (hypertension, diabetes, smoking, and prior myocardial infarction). Considering these results, we propose that Aβ-42 can add value to risk stratification in NSTEMI patients.

## 1. Introduction

Ischemic heart disease is considered the leading cause of mortality worldwide and the main cause of disease burden in developed countries, accounting for nearly 9.4 million deaths in 2021 [1,2]. Ischemic heart disease from a pathophysiological standpoint is a consequence of reduced blood flow from the coronary arteries that supply the energetic necessities of the heart. Reduced blood flow is often caused by coronary atherosclerosis; however, other conditions such as coronary microvascular dysfunction, inflammation, and vasospasm may precipitate myocardial ischemia. Myocardial infarction is the irreversible necrosis of cardiac myocytes due to extended ischemia by the deprivation of oxygen supply and/or inflammatory, metabolic, or toxic insults [3,4,5]. Acute coronary syndromes encompass a spectrum of conditions that include ST elevation myocardial infarction (STEMI), non-ST elevation myocardial infarction (NSTEMI), and unstable angina (UA) [6,7]. Coronary thrombus development in vulnerable atherosclerotic plaque is the primary cause of acute coronary syndrome; nevertheless, a considerable number of patients undergoing ACS may be caused by plaque erosion, calcific nodules, coronary spasms, and spontaneous coronary artery dissection [8,9].

Beta amyloid (Aβ) is a peptide with a length of 37–49 amino acids that are produced by the excision of the amyloid precursor peptide (APP). APP can be cleaved by α -secretase (nonamyloidogenic pathway) or β -secretase (amyloidogenic pathway) to produce α or β-C terminal fragments, respectively. The additional breakdown of β-C terminal fragments by γ -secretase can release different cleavage fragments that are further processed into the main isoforms of Aβ, the 40 amino acid Aβ-40 and the 42 amino acid Aβ-42 [10,11]. These primary isoforms are found in vascular lesions and parenchymal lesions in the brain [12]. An experimental study has demonstrated that an early peak of soluble amyloid precursor peptide (sAPP) preceded the liberation of myocardial injury enzymes [13]. There is evidence of the participation of Aβ in thrombosis [14] and clinical manifestations of acute coronary syndrome [15]. Furthermore, Aβ-40 stimulates the activation and adhesion of platelets [16,17,18,19], partially by the PLCγ2-PKC pathway; in addition, platelet aggregation may be further amplified by thromboxane A2 formation and ADP secretion. Aβ-40 can also induce monocytes to release MMP-9, a matrix metalloproteinase capable of increasing the vulnerability of the plaque by disrupting its basement membrane and promoting the infiltration of monocytes/macrophages into atherosclerotic lesions [20,21]. Moreover, Aβ-42 affects coronary endothelial cells and cardiomyocytes, reducing mitochondrial respiration and disruption of fatty acid metabolism in both cell types [22]. However, there are limited studies that addressed the circulating levels of these peptides and conventional biomarkers in ST elevation and non-ST myocardial infarction. In this study, we investigated the correlation of Aβ peptides with myocardial injury and inflammation biomarkers in acute coronary syndrome.

## 2. Materials and Methods

### 2.1. Study Population

This single-center, cross-sectional, observational, and correlation study included patients within the first 12 h after the onset of symptoms (chest pain, shoulder pain, or chest discomfort) with acute coronary syndrome who were admitted to the coronary care unit at the Instituto Nacional de Cardiología Ignacio Chávez from Mexico City. Patients’ exclusion criteria were as follows: patients with a history of renal disease, liver failure, autoimmune or autoinflammatory disease, and malignant or hematological disorders. Patients with inadequate blood sample volume to evaluate Aβ peptide (Aβ-42, Aβ-40) concentration, a period of more than 180 min between the collection of blood sample and storage at −80 °C, or those who wish to withdraw consent were eliminated. 

The Institutional Research and Ethics committees approved this study (protocol number 21-1275) in compliance with the principles outlined in the Helsinki Declaration. Informed oral and written consent was given by all the subjects participating in this study.

### 2.2. Sampling Method and Sample Size Determination 

A non-probabilistic sampling method (convenience method) was used to select all consecutive patients who fulfilled the selection criteria described above. There are no previous studies on the association between the plasmatic levels of Aβ peptides (Aβ-42 and Aβ-40) and biomarkers of myocardial injury or inflammation in patients with acute coronary syndrome. Therefore, we proposed an expected minimal Pearson correlation coefficient between Aβ peptides (Aβ-42, Aβ-40) and myocardial injury or inflammation biomarkers of 0.3. Assuming a one-tail alfa error of 0.05 and a beta error of 0.2 (i.e., a statistical power of 0.8), we calculated a necessary sample size of 68 subjects using the computer program G*Power version 3.1.9.6 [23].

### 2.3. Data Collection

The patients’ hospital charts and electronic medical history were reviewed to obtain all clinical data from patients. Clinical characteristics included age, sex, body mass index (kg/m), presence of diabetes, hypertension, dyslipidemia, previous myocardial infarction, smoking status, Killip–Kimball class, Global Registry of Acute Coronary Events (GRACE) score, Thrombolysis In Myocardial Infarction (TIMI) score for ST elevation myocardial infarction (STEMI), TIMI score for unstable angina/non-ST elevation myocardial infarction (UA/NSTEMI) score, Can Rapid risk stratification of Unstable angina patients Suppress ADverse outcome with Early implementation of the American College of Cardiology/American Heart Association Guidelines (CRUSADE) score, and left ventricular ejection fraction (LVEF %). We also gathered time parameters (minutes), such as symptom-to-door, symptom-to-blood sample (time between symptom onset and blood sample collection), door-to-electrocardiogram (ECG), door-to-needle, door-to-balloon, and symptom-to-catheter. 

Biochemical values obtained from the electronic medical history included high-sensitivity troponin I (hs-cTnI [pg/mL]), N-terminal pro-B natriuretic peptide (NT-proBNP [pg/mL]), high-sensitivity C-reactive protein (hs-CRP [mg/dL]), total cholesterol (mg/dL), high-density lipoprotein (HDL [mg/dL]), low-density lipoprotein (LDL [mg/dL]), triglycerides (mg/dL), atherogenic index of plasma (AIP), and albumin (g/mL). In addition, we quantified plasmatic levels of Aβ-42 (pg/mL), Aβ-40 (pg/mL), and Aβ-42/40 ratio from a blood sample. 

### 2.4. Assessment of Plasmatic Aβ Peptides (Aβ-42, Aβ-40) 

Peripheral blood from subjects was obtained in a 4 mL BD K2EDTA Vacutainer tube. The blood was then centrifuged (1500 rpm, 15 min at 4 °C) and plasma was collected and stored in 200 µL aliquots at −80 °C until Aβ peptide analysis. 

After thawing at room temperature, plasmas were used to measure Aβ-42 and Aβ-40 levels with a High-Sensitivity Human Amyloid Aβ-42 ELISA and a High-Sensitivity Human Amyloid Aβ-40 ELISA Kit (both from EMD Millipore Corporation, Billerica, MA, USA) following the manufacturer’s instructions. Absorbance measurements were taken using an imaging reader (BioTek Cytation 3 Cell Imaging Multimode Reader, Agilent, CA, USA) at 450 nm and 590 nm. The measurement range was 16–500 pg/mL for Aβ-42 and Aβ-40. The intra- and inter-assay coefficients of variation of ELISA kits were <10%. No cross-reactivity was observed between Aβ-42 and Aβ-40 antibodies.

### 2.5. Statistical Methods

Categorical variables are presented as frequencies and percentages. Comparisons were made using the chi-squared test or Fisher’s exact test. The normal distribution of continuous variables was tested using the Kolmogorov–Smirnov method. Continuous variables with a normal distribution are described as mean ± SD and were compared using Student’s *t*-test. Otherwise, the Mann–Whitney’s U test was used, and variables were described as median (percentile 25–percentile 75). We calculated the correlation between Aβ peptides (Aβ-42, Aβ-40, and Aβ-42/40 ratio) and continuous variables with the Spearman correlation test since none of them had a normal distribution. Receiver operator characteristic (ROC) curve analysis was performed to evaluate the performance of Aβ-42 to distinguish between patients from the STEMI and NSTEMI groups. The shortest orthogonal distance between the ROC curve points and the optimum point (0,1) was calculated to identify the best cut-off value. A multivariate binary regression analysis was applied to assess the independent association of Aβ-42 with the type of myocardial infarction (NSTEMI/STEMI) as the dependent variable when considering Aβ-42, diabetes, hypertension, previous myocardial infarction, smoking status, and the number of SMuRFs as independent variables. The statistical analysis was performed using Statistical Package for the Social Sciences (SPSS) version 21.0 (IBM Corp., Armonk, NY, USA). A *p*-value < 0.05 was considered statistically significant. 

## 3. Results

### 3.1. Overall Patient Characteristics and Demographics

The study sample included 65 patients, with a mean age of 59 ± 13 years. Fifty-one patients (78.5%) were male. Fifty-nine patients (90.8%) had at least one SMuRF (standard modifiable cardiovascular risk factor). The number of patients with risk factors were arterial hypertension, 42 (64.6%), diabetes mellitus, 14 (21.5%), dyslipidemia, 12 (18.5%), smoking, 44 (67.7%), and previous myocardial infarction, 27 (41.5%). 

Fourteen patients (21.5%) had a Killip–Kimball of two and above class, forty (61.53%) had an intermediate–high risk GRACE score, forty-two (64.61%) had an intermediate–high risk TIMI score, twenty-nine (44.6%) had a moderate–high risk CRUSADE score, nineteen (30.6%) patients had a mid-range LVEF (left ventricular ejection fraction), and fifteen (24.2%) had reduced a LVEF. The mean symptom-to-door time was 339 ± 191 min, symptom-to-blood sample time was 390 ± 194 min, and symptom-to-catheter time was 2147 ± 2203 min. 

Biochemical levels of Aβ-42 were 38.39 (35.39–43.63) pg/mL and Aβ-40 176.20 ± 79.35 pg/mL, the Aβ-42/40 ratio was 0.25 (0.20–0.31), hs-cTnI was 333.0 (67.10–1617.0) pg/mL, NT-proBNP was 342.0 (111.0–975.0), hs-CRP was 4.57 (2.26–9.50) mg/dL, albumin was 4.04 ± 0.48 g/dL, total cholesterol was 165.0 ± 40.7 mg/dL, triglycerides was 157.1 ± 63.3 mg/dL, HDL was 37.6 ± 8.3 mg/dL, LDL was 106.5 ± 38.1 mg/dL, and AIP was 0.23 ± 0.21.

### 3.2. Aβ Peptide Correlation Analysis

We analyzed the data recovered from our constructed database, and Spearman’s correlation test was applied. Table 1 shows the correlation between the Aβ-42, Aβ-40, and Aβ-42/40 ratio with our study variables. We found a positive correlation between Aβ-42 and Aβ-40 with age, Aβ-42 and Aβ-40 with NT-proBNP, and Aβ-40 with ≥1 SMuRF. Conversely, we found a negative correlation between Aβ-42 and albumin and the Aβ-42/40 ratio with ≥1 SMuRF (ρ = −0.251, *p* = 0.044). There was no correlation with statistical significance between Aβ-42, Aβ-40, and the Aβ-42/40 ratio and other biochemical variables, such as hs-cTnI, hs-CRP, albumin, total cholesterol, HDL, LDL, triglycerides, and AIP. 

Figure 1 provides scatterplots with marginal histograms to represent visually the correlation of Aβ-42, Aβ-40, and Aβ-42/40 with age, NT-proBNP, and albumin. At first glance, we observed a right skewness in the distribution histograms of Aβ-42, Aβ-40, Aβ-42/40 ratio, and NT-proBNP; additionally, in the scatterplots of these variables, some data points were extreme values regarding the main distribution and are considered outliers.

Considering the potential heightened effect of these outliers in the correlation between our study variables, we established cut-off values and repeated the correlation analysis. Outliers were defined by Aβ-42 < 20 pg/mL or >100 pg/mL, Aβ-40 > 400 pg/mL, the Aβ-42/40 ratio < 1.0, and NT-proBNP > 10,000 pg/mL. After this additional analysis, the positive correlation between Aβ-40 with age, Aβ-42, and Aβ-40 with NT-proBNP was maintained; in addition, we observed a negative correlation between the Aβ-42/40 ratio with age. The rest of the correlations were not statistically significant. This analysis is available in the Appendix A.

### 3.3. Comparison of Aβ-42 and NT-proBNP with Cardiovascular Risk Factors

Table 2 shows the mean levels of plasmatic Aβ-42 and NT-proBNP between several cardiovascular risk factors, including diabetes, hypertension, dyslipidemia, smoking, prior MI, and two or more SMuRFs. We observed higher levels of Aβ-42 in patients with hypertension compared to those without hypertension, and plasmatic levels of Aβ-42 had a tendency to be higher in patients with diabetes and prior MI when compared to patients without these risk factors. Conversely, Aβ-42 was higher in nonsmokers without dyslipidemia when compared to those who smoked and had dyslipidemia. 

### 3.4. Comparison of Demographic, Clinical, and Biochemical Characteristics between Groups

Table 3 presents the demographic and laboratory baseline characteristics. There were 34 patients in the STEMI group (52%) and 31 patients in the NSTEMI group (47%). We observed a higher prevalence of diabetes, hypertension, smoking, and previous myocardial infarction in the NSTEMI group compared to the STEMI group. When repeating this analysis without outliers, the higher prevalence of hypertension (*p* = 0.046), previous MI (*p* = 0.01), and smoking (*p* = 0.005) in the NSTEMI group when compared to the STEMI group was preserved. These data are available in the Appendix A.

We observed no difference in hs-cTnI, NT-proBNP, hs-CRP, albumin, total cholesterol, triglycerides, HDL, LDL, or AIP between our study groups (Table 4). 

Regarding Aβ peptides, we observed (Figure 2) that circulating levels of Aβ-42 were higher in the NSTEMI group compared to the STEMI group (median level 41.76 [38.99–47.75] vs. 35.96 pg/mL [34.27–39.14]; *p* = 0.001). There was no evident difference in Aβ-40 and the Aβ-42/Aβ-40 ratio between both groups (Appendix A). Interestingly, when repeating this analysis without outliers (Appendix A), the plasmatic levels of Aβ-42 were higher in the NSTEMI group compared to the STEMI group (median level 40.26 [37.64–45.13] vs. 35.39 pg/mL [34.27–37.64]; *p* = 0.001), and no difference between Aβ-42 and the Aβ-42/40 ratio when compared in both groups was found. 

The ROC curve analysis in Figure 3 shows that Aβ-42 distinguishes between patients in the STEMI and NSTEMI groups. The cut-off value (Aβ-42 ≥ 38 pg/mL) was estimated as the best to distinguish NSTEMI patients from STEMI patients with the whole sample (sensitivity = 77%, specificity = 74%) and the sample without outliers (sensitivity = 74%, specificity = 80%).

Table 5 shows that having elevated Aβ-42, i.e., Aβ-42 ≥ 38 pg/mL, was independently associated with NSTEMI. The model tested with all patients showed that patients with elevated Aβ-42 had over 12 times higher risk of having NSTEMI, while the risk of having NSTEMI was even higher in the model tested without outliers. 

## 4. Discussion

The traditional classification of acute myocardial infarction (AMI) based on electrocardiogram findings (STEMI or NSTEMI) for the patient’s initial management has led to a more guided approach to the different clinical presentations in the spectrum of acute coronary syndrome [7]. Even though STEMI and NSTEMI have been associated with similar pathophysiologic mechanisms sharing standard modifiable cardiovascular risk factors (SMuRFs) [24], they show different prevalence of these risk factors, including age. 

Aβ peptides have been associated with peripheral atherosclerotic manifestations [25], vascular inflammation [12], myocardial dysfunction [26,27], mortality, and adverse cardiovascular outcomes [28,29]. Aβ-40 is related to subclinical cardiovascular disease and has been used for risk stratification in NSTE-ACS [27,28,29,30]. However, there are no studies on the role of Aβ-42 in STEMI and/or NSTEMI. We aimed to investigate the correlation between plasmatic levels of beta-amyloid peptides (Aβ-42, Aβ-40, Aβ-42/40 ratio) with myocardial injury and inflammatory biomarkers in patients with acute coronary syndrome. 

Our results showed that NSTEMI patients had a different prevalence of risk factors and Aβ levels. We observed a greater prevalence of hypertension, diabetes, and previous myocardial infarction in NSTEMI when compared to STEMI. These findings were consistent with previous studies performed in the US and Europe [31,32,33]. However, we also found that smoking was more prevalent in NSTEMI patients, opposing the previously reported higher prevalence of smoking in STEMI patients. This difference could be explained because of the inclusion of former smokers in our study as part of the categorization of smoking status. The National Health Interview Survey defined former smokers as adults who had smoked at least 100 cigarettes in his or her lifetime but who had quit smoking at the time of the interview [34]. This additional subcategory was not clear in the aforementioned studies, so further research regarding the impact of former smokers as a risk factor is necessary. 

When comparing Aβ peptides (Aβ-40, Aβ-42) in patients with acute coronary syndrome, we found higher levels of Aβ-42 in the NSTEMI group. Aβ-42 levels can distinguish NSTEMI from STEMI patients with sensitivity and specificity above 70%, and the association of elevated Aβ-42 with a higher risk of having NSTEMI was independent of other NSTEMI risk factors (diabetes, hypertension, prior myocardial infarction, and smoking status). The cleavage of APP in endothelial cells [13], macrophages [35], platelets [17], and neurons [36] may be influenced by factors such as aging, ischemia, and inflammation [12]. This enhancement of APP/Aβ processing and inadequate clearance may favor the accumulation of Aβ peptides in the bloodstream, vascular wall [12], and heart tissue [37]. Notably, in previous studies, STEMI patients had a higher mortality rate compared to NSTEMI, but this tendency changes at one- or two-year follow-up when mortality rates become similar between both groups [24]. This may be due to the greater prevalence of cardiovascular risk factors in NSTEMI and could be correlated with enhancing the APP/Aβ processing and residual inflammation over time. To our knowledge, there is no evidence of what could modulate the final processing of APP to generate pathologic cleavage products and subsequently an inclination towards a specific length of Aβ (Aβ-40 or Aβ-42) in each of the different cell types. Platelets have great importance in releasing circulating Aβ peptides [17] along with different pathways of activation in STEMI and NSTEMI, favoring thromboxane receptor and PAR1 pathways, respectively [38]. A recent report demonstrated that human platelets release higher levels of Aβ-42 from α granules in response to the combination of hypoxia and inflammation [39]. In myocardial infarction, platelet activation precedes coronary thrombosis. Therefore, further studies exploring the production or storage of Aβ-42 in these phenotypically different platelets may add value to the usage of Aβ peptides in risk stratification. 

We also found a correlation between age and NT-proBNP with both Aβ peptides (Aβ-40, Aβ-42), but albumin only correlates with Aβ-42. Aβ peptide production and clearance can be affected by age-related mechanisms. Neprilysin (NEP) is a metallo-endopeptidase that degrades several bioactive peptides, including Aβ peptides [40], and there is evidence that supports the hypothesis that aging can reduce NEP activity, thereby leading to Aβ accumulation [11]. Furthermore, neuronal aging can increase APP endocytosis, consequently enhancing Aβ production [41]. A study reported that plasmatic levels of Aβ-42 in cognitively and neurologically normal individuals increase with age. However, these levels stabilize after age 65 [42]. With the additional correlation analysis, by excluding potential outliers, only the correlation between Aβ-40 and age was conserved.

The use of circulating natriuretic peptides (NPs), including BNP and NT-proBNP, as clinical biomarkers revolutionized the early recognition of patients with heart failure and ruled out other causes of dyspnea [43], as well as risk stratification after acute myocardial infarction [44]. NPs are secreted by cardiomyocytes through different pathways and are mainly stimulated by myocardial stretching, neurohormones (endothelin 1 and angiotensin II), and circulating cytokines (IL-1β or TNF), which involve G_o_α, G_q_α, or p38 activation [40]. The novel finding of this study is the association of Aβ-42 with NT-proBNP in patients with ongoing acute coronary syndrome. Focusing research on Aβ-42 in acute coronary syndrome in further studies could help explore the underlying participation of this peptide in cardiovascular disease. A study found that plasma Aβ-40 was associated with NT-proBNP, suggesting that Aβ-40 could be involved in early subclinical rise in filling pressure in the general population without overt coronary cardiovascular disease [27]. Moreover, circulating Aβ-40 is a predictor of mortality and can improve risk stratification of patients with NST-ACS over GRACE score [30]. 

Our results show a negative correlation between Aβ-42 and albumin. Serum albumin has an important role in balancing Aβ peptides between brain and blood plasma due to its capability of binding. A study found that low serum albumin may increase amyloid accumulation in patients with Alzheimer’s Disease [45]. Although, the regulation of Aβ peptides by albumin in cardiovascular disease has not yet been investigated. 

On the other hand, we did not find a correlation between Aβ peptides (Aβ-40, Aβ-42) and high-sensitivity troponin I (hs-cTnI). Cardiac troponin I can be detected in serum early after the onset of acute myocardial infarction, and usually peak levels are reached after 12–48 h [46]. Therefore, the timing of the blood sample could have an influence on this association. A study explored the changes in cardiac troponins I and T beyond the initial hours of symptom onset. They found a peak concentration of 6–12 h for cTnI and 12–18 h for cTnT from initial sampling. However, the time between symptom onset and initial sampling is not clear [47]. 

Finally, there was no correlation between Aβ-42 and high-sensitivity C-reactive protein (hs-CRP) in our population. There is limited information regarding this with ongoing acute coronary syndrome. However, a previous study reported no association between higher levels of CRP and Aβ-42 in cerebral small vessel disease [48].

Our study has some limitations. Firstly, we estimated a population size sample of 68 subjects, which is relatively small and will need to be applied to a larger cohort. Secondly, our study groups were predominantly male and older age adults. Therefore, our findings may need further validation and should not be generalized in female subjects and younger populations. Thirdly, due to the cross-sectional nature of our study, we only assessed Aβ peptide, cardiac injury, and inflammatory biomarkers at the admission of patients who arrived in the first 12 h of symptom onset, although serial determinations of these biomarkers were not part of our objective, further studies regarding the kinetics of these biomarkers are needed. Nevertheless, our results were the first to explore plasmatic levels Aβ-42 with conventional biomarkers in acute coronary syndrome. Additional longitudinal studies are required to determine whether the association of NT-proBNP and both Aβ peptides (Aβ-40, Aβ-42), together with additional echocardiographic measurements, may provide insight into heart failure after myocardial infarction.

## 5. Conclusions

The present study demonstrates the correlation between Aβ-42 and NT-proBNP in patients with ST elevation and non-ST elevation myocardial infarction. The plasmatic levels of Aβ-42 are higher in NSTEMI when compared to STEMI. Considering the toxic properties of Aβ-42 in coronary endothelial cells and cardiomyocytes, this peptide may be useful in risk stratification scores in patients with NSTEMI. Further longitudinal studies may clarify the role of Aβ-42 in high-risk patients with ongoing acute coronary syndrome. 

## Figures and Tables

**Figure 1 jcm-13-01117-f001:**
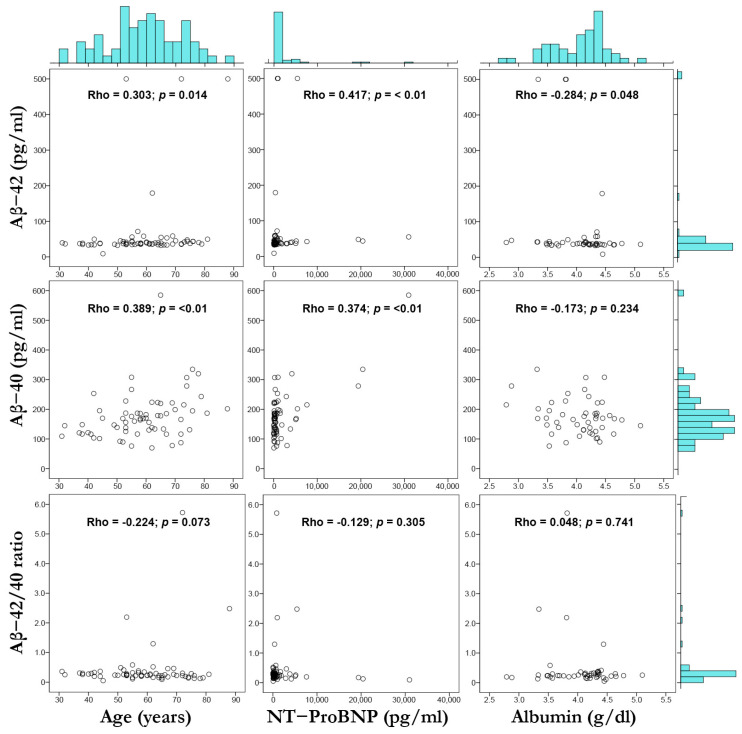
Scatterplot matrix and marginal histograms (with Spearman’s correlation coefficients) showing the relation between Aβ-42 vs. age, NT-proBNP, and albumin (**top row**); Aβ-40 vs. age, NT-proBNP, and albumin (**middle row**); Aβ-42/40 ratio vs. age, NT-proBNP, and albumin (**bottom row**). Spearman’s correlation (Rho) and *p* value are shown.

**Figure 2 jcm-13-01117-f002:**
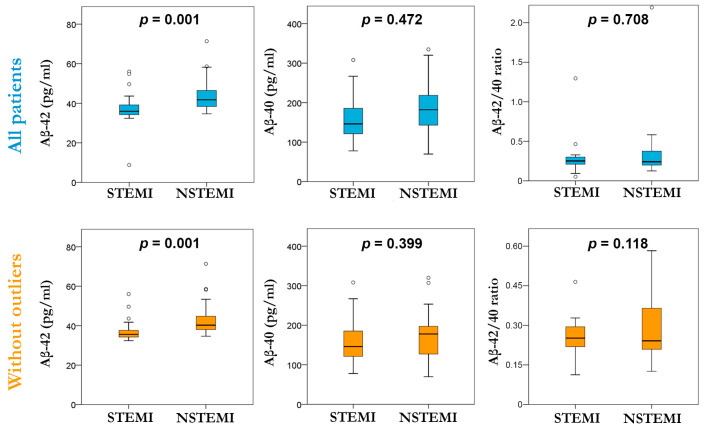
Box plot of plasmatic levels of Aβ-42, Aβ-40, and the Aβ-42/40 ratio evaluated in the STEMI and NSTEMI groups considering all patients (**top row**) and without outliers (**bottom row**).

**Figure 3 jcm-13-01117-f003:**
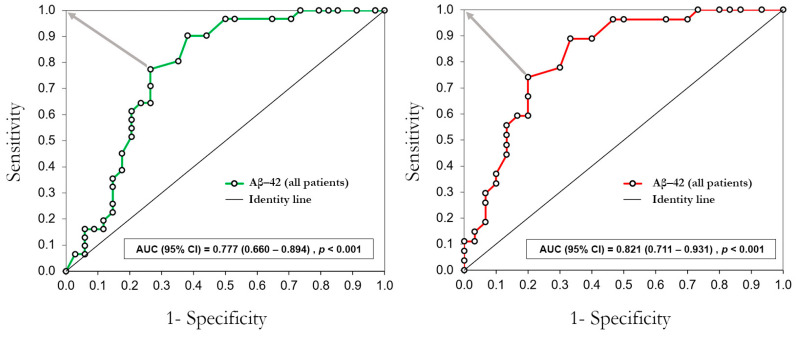
ROC curve analysis of plasmatic levels of Aβ-42 to distinguish patients in the NSTEMI group and the STEMI group. The gray arrows indicate the shortest orthogonal distance between the ROC curve points and the optimum value (0,1). AUC = area under the curve; CI = confidence interval.

**Table 1 jcm-13-01117-t001:** The correlation between our study variables and Aβ peptides (Aβ-42, Aβ-40) and the Aβ-42/40 ratio in patients with acute coronary syndrome. Data were analyzed using Spearman’s correlation test.

Variable	Aβ-42 (pg/mL)		Aβ-40 (pg/mL)		Aβ-42/40 Ratio	
	Rho	*p*-Value	Rho	*p*-Value	Rho	*p*-Value
Age (years)	0.303	0.014	0.389	0.001	−0.224	0.073
BMI (kg/m^2^)	−0.050	0.694	−0.084	0.507	−0.031	0.805
One or more SMuRF	0.167	0.184	0.298	0.016	−0.251	0.044
High-sensitivity troponin I (pg/mL)	−0.021	0.869	0.063	0.617	−0.142	0.260
CRP (mg/dL)	0.073	0.584	0.174	0.190	−0.155	0.247
NT-proBNP	0.417	0.001	0.374	0.002	−0.129	0.305
Albumin (g/dL)	−0.284	0.048	−0.173	0.234	0.048	0.741
Total cholesterol (mg/dL)	−0.153	0.274	−0.089	0.526	0.061	0.664
HDL (mg/dL)	−0.005	0.972	0.061	0.657	−0.043	0.757
LDL (mg/dL)	−0.147	0.285	−0.096	0.484	0.090	0.515
Triglycerides	−0.074	0.611	−0.051	0.726	−0.051	0.725
AIP	−0.073	0.617	−0.079	0.585	−0.024	0.869
Symptom-to-blood sample ^†^	0.012	0.923	−0.142	0.259	0.049	0.697

^†^ Time between symptom onset and blood sample collection.

**Table 2 jcm-13-01117-t002:** Aβ-42 and NT-proBNP compared by the presence of cardiovascular risk factors, including diabetes, hypertension, dyslipidemia, smoking, and prior myocardial infarction. Data are shown as median (percentile 25–percentile 75).

Variable	All Subjects (*n* = 65)			
	Aβ-42 (pg/mL)	*p*-Value	NT-proBNP (pg/mL)	*p*-Value
Diabetes				
Yes: No:	42.50 (36.51–47.75) 37.64 (35.01–43.63)	0.075	3978.0 (226.0–7611.0) 277.0 (110.0–690.0)	0.004
Hypertension				
Yes: No:	39.70 (36.51–49.62) 36.14 (34.64–39.51)	0.006	551.0 (181.0–1862.0) 231.0 (110.0–522.0)	0.056
Dyslipidemia				
Yes: No:	37.07 (35.58–39.13) 38.76 (35.39–44.38)	0.393	415.5 (76.7–1192.5) 342.0 (125.0–974.0)	0.819
Smoking				
Yes: No:	38.39 (35.39–43.25) 38.57 (35.76–48.68)	0.624	226.0 (99.0–975.0) 642.5 (257.5–1125.0)	0.060
Prior MI				
Yes: No:	40.26 (36.14–53.37) 37.45 (35.01–41.76)	0.066	516.0 (199.0–1862.0) 257.5 (110.0–766.0)	0.197
SMuRF				
≥One: None:	38.76 (35.76–45.13) 35.01 (33.52–37.64)	0.039	315.0 (110.0–978.0) 470.0 (234.0–642.0)	0.910

**Table 3 jcm-13-01117-t003:** Demographic characteristics of 65 patients admitted to the coronary care unit grouped by ST elevation myocardial infarction (STEMI) or non-ST elevation myocardial infarction (NSTEMI). Data are shown as mean ± standard, median (percentile 25–percentile 75), and absolute value (percentage).

Variable	STEMI (*n* = 34)	NSTEMI (*n* = 31)	*p*-Value
Age (years)	58 ± 12	60 ± 12	0.503
Sex			
Male Female	29 (85.3%) 5 (14.7%)	22 (71.1%) 9 (29.9%)	0.135
Body mass index (kg/m^2^) Overweight Obesity Overweight/Obesity	28.28 ± 4.15 14 (41.2%) 12 (35.3%) 26 (76.5%)	27.83 ± 3.70 14 (45.2%) 9 (29.0%) 23 (74.0%)	0.652
Diabetes	4 (11.8%)	10 (32.3%)	0.043
Hypertension	18 (52.9%)	24 (77.4%)	0.039
Dyslipidemia	6 (17.6%)	6 (19.4%)	0.859
Previous myocardial infarction	9 (26.5%)	18 (58.1%)	0.010
Smoking status			
Current smoker Former smoker Nonsmoker	16 (47.1%) 6 (17.6%) 12 (35.3%)	5 (16.1%) 17 (54.8%) 9 (29.1%)	0.003
Number of SMuRFs			
One or more None	29 (85.3%) 5 (14.7%)	30 (96.3%) 1 (3.7%)	0.121
Killip–Kimball			
≥2 Class	9 (26.5%)	5 (16.1%)	0.240
GRACE			
Intermediate–High Risk	22 (64.7%)	18 (58.1%)	0.583
TIMI			
Intermediate–High Risk	21 (61.8%)	21 (67.7%)	0.615
CRUSADE			
Moderate–High Risk	14 (41.2%)	15 (48.4%)	0.559
LVEF			
Mid-range ejection fraction Reduced ejection fraction	9 (27.3%) 11 (33.3%)	6 (20.7%) 8 (27.6%)	0.618
Symptom-to-door time (minutes)	392 ± 196	280 ± 171	0.018
Symptom-to-blood sample time (minutes)	418 ± 198	364 ± 193	0.221
Door-to-electrocardiogram time (minutes)	6 (5–8)	5 (5–7)	0.143
Door-to-balloon time (minutes)	508 (98–2377) (*n* = 31)	2030 (1085–3576) (*n* = 16)	0.001
Symptom-to-catheter time (minutes)	1700 ± 1889	3070 ± 2562	0.992

**Table 4 jcm-13-01117-t004:** Biochemical values from 65 patients admitted to the coronary care unit grouped by ST elevation myocardial infarction (STEMI) or non-ST elevation myocardial infarction (NSTEMI). Data are shown as mean ± standard deviation or median (percentile 25–percentile 75).

Variable	STEMI (*n* = 34)	NSTEMI (*n* = 31)	*p*-Value
Aβ-42 (pg/mL)	35.96 (34.27–39.14)	41.76 (38.99–47.75)	0.001
Aβ-40 (pg/mL)	169.38 ± 88.26	183.68 ± 68.96	0.472
Aβ-42/40 ratio	0.25 (0.21–0.30)	0.24 (0.20–0.38)	0.708
High-sensitivity troponin I (pg/mL)	368.5 (67.1–4742.0)	249 (65.3–1299.0)	0.281
NT-proBNP (pg/mL)	257.5 (104.0–643.0)	516.0 (199.0–643.0)	0.097
High-sensitivity CRP (mg/dL)	4.98 (2.47–9.50)	4.46 (1.63–10.40)	0.569
Albumin (g/dL)	4.14 ± 0.42	3.95 ± 0.52	0.149
Total cholesterol (mg/dL)	170.3 ± 43.7	157.5 ± 35.7	0.264
Triglycerides (mg/dL)	162.2 ± 65.8	149.4 ± 60.3	0.490
HDL (mg/dL)	38.65 ± 8.81	36.21 ± 7.65	0.285
LDL (mg/dL)	112.9 ± 41.4	98.1 ± 32.3	0.155
AIP	0.23 ± 0.23	0.23 ± 0.20	0.992

**Table 5 jcm-13-01117-t005:** Multivariate logistic regression analysis of the association between the type of myocardial infarction (STEMI or NSTEMI) as a dependent variable and Aβ-42, diabetes, hypertension, previous MI, smoking status, and the number of SMuRFs as independent variables. Odds ratio (OR) are shown as absolute values.

Variable	All Patients (*n* = 65)			Without Outliers (*n* = 57)		
	β Coefficient	OR (95% CI)	*p*-Value	β Coefficient	OR (95% CI)	*p*-Value
Aβ-42 ≥ 38 pg/mL	2.579	13.184 (2.943–59.053)	0.001	3.672	39.313 (4.479–345.063)	0.001
Diabetes	1.593	4.918 (0.820–29.510)	0.081	1.996	7.361 (0.714–75.859)	0.093
Hypertension	−0.696	0.499 (0.090–2.753)	0.425	−0.795	0.451 (0.071–2.856)	0.398
Previous myocardial infarction	1.292	3.639 (0.763–17.351)	2.131	8.424	8.424 (1.087–65.268)	0.041
Smoking status * Former smoker Nonsmoker	1.582 −0.916	4.863 (0.915–25.838) 0.400 (0.083–1.940)	0.063 0.256	2.333 −1.004	10.308 (1.210–87.783) 0.366 (0.056–2.427)	0.033 0.298

* Current smoker is the reference category.

## Data Availability

The raw data supporting the conclusions of this article will be made available upon request to the corresponding author, provided pertinent legal requirements are met.

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
