# Peer review of "Correlation of Plasmatic Amyloid Beta Peptides (Aβ-40, Aβ-42) with Myocardial Injury and Inflammatory Biomarkers in Acute Coronary Syndrome"

_jcm, 2024, doi:10.3390/jcm13041117_

Round 1

Reviewer 1 Report

Comments and Suggestions for Authors

Dear Authors,

I would like to state that I read the article with interest. But although it seems good in general terms, it is obvious that there are some shortcomings. Because;

1- Abstract: Adding the start and end date of the article, age, gender ratios, and actual p values of the results will strengthen the summary.

2- Introduction: needs to be strengthened further.

3- Material and Method: * Acceptance and rejection criteria should be written clearly. According to what criteria did the authors exclude the diseases they excluded?

                   * The number of cases included in the study is very small and in such cases, it does not reflect the general statistical studies. By what method was the number of cases determined? The universe size, margin of error, and acceptability rate should be explained.

                 * A simple statistical method was used in the study. However, for the study to be more powerful, regression analysis, univariate and multivariate analysis, and ROC curve analysis should have been performed. The statistical analysis needs to be redone.

  4- The discussion should be re-evaluated according to the new statistical data.

   Kind regards.

Comments on the Quality of English Language

There are grammatical and language explosions.

Author Response

Comment 1: Dear Authors, I would like to state that I read the article with interest. But although it seems good in general terms, it is obvious that there are some shortcomings. Because;

1- Abstract: Adding the start and end date of the article, age, gender ratios, and actual p values of the results will strengthen the summary.

Response: Abstract was re-written, data of demographic characteristics and values p were added.

Comment 2: 2- Introduction: needs to be strengthened further.

Response:  In order to strengthen the introduction, we defined ischemic heart disease emphasizing the main causes involved in its pathophysiology. Amyloid peptide processing pathways were described to elucidate how the main isoforms involved in cardiovascular disease are synthesized. Evidence supporting the participation of Aβ-40 in atherosclerosis was added.

Comment 3: 3- Material and Method: * Acceptance and rejection criteria should be written clearly. According to what criteria did the authors exclude the diseases they excluded?

Response: Acceptance criteria were more described in Methods. Patients with  history of renal disease, liver failure, autoimmune or autoinflammatory disease, and malignant or hematological disorders were excluded because specific metabolic characteristics of each entity can influence the APP processing and consequently modified AB40 and AB42 levels.

Comment 4: * The number of cases included in the study is very small and in such cases, it does not reflect the general statistical studies. By what method was the number of cases determined? The universe size, margin of error, and acceptability rate should be explained.

The discussion should be re-evaluated according to the new statistical data.

Response: We have updated the Discussion section accordingly (lines 292 to 295).

We agree with the reviewer about the relatively small sample size, as it is mentioned in the study limitations (page 12, lines 359-360). In section 2.2 (page 2), we describe how we estimated the sample size considering an expected minimum Pearson correlation of 0.3. We assumed a one-tail alfa error of 0.05 and a beta error of 0.2 (i.e., statistical power of 0.8). We calculated the sample size with the computer program G*Power version 3.1.9.6. The description has been updated, and the reference describing the method used by the computer program was also added. 

We did not consider a universe size or acceptability rate for this calculation since our sampling method was non-probabilistic. The sampling method is now described in lines 74-75.

Comment 5:  * A simple statistical method was used in the study. However, for the study to be more powerful, regression analysis, univariate and multivariate analysis, and ROC curve analysis should have been performed. The statistical analysis needs to be redone.

Response: We updated the analysis by adding ROC curve analysis to evaluate the performance of Aβ-42 to distinguish between patients from the STEMI and NSTEMI groups, and multivariate binary logistic regression analyses to assess the independent association of Aβ-42 with the type of myocardial infarction (NSTEMI/STEMI), as is now described in section 2.4 Statistical Methods (lines 126 to 134). The corresponding results are in the new Figure 2 (lines 230 to 239) and Table 5 (lines 240 to 247).

Regarding the analysis for the main objective (i.e., to assess the correlation of Aβ peptides with myocardial injury and inflammation biomarkers in acute coronary syndrome), we only performed univariate and bivariate analysis because the Aβ peptides had no normal distribution (as mentioned in line 125, of the Statistical methods section and described on section 3.2, “Aβ peptides correlation analysis”). The distribution of Aβ peptides was very skewed, as described in Figure 1. Even after testing the distributions without the outlier values or transforming the data points using logarithmic functions, the data had no normal distribution. Hence, we could use only non-parametric bivariate analysis (Spearman correlations) since multivariate linear regression analyses would not be appropriate for this data.

Reviewer 2 Report

Comments and Suggestions for Authors

In this cross sectional study authors analyzed correlation of plasma Aß-40 and Aß-42 with markers of myocardial injury and inflammation in patients with STEMI and NSTEMI.

Technically the manuscript is well written- methodology is good, although there is a small number of analyzed patients. Results are clearly presented. Conclusions are in concordance with findings.

My comments are:

  1. Authors analyzed correlation between Aß-40 and Aß-42 and cholesterol, HDL, LDL. These are not biomarkers of myocardial injury nor inflammatory biomarkers, so in my opinion correlation with these biomarkers is not of clinical importance.

  2. Table 2. NYHA class- in patients with acute coronary syndrome we usually classify heart failure according to Killip-Kimball class only.

  3. Authors stated that elevation of  Aß-40 and Aß-42 in serum precede the liberation of myocardial injury enzymes. Since hs-troponin assay can detect elevated troponin level one hour after the symptom onset. When can elevated levels of Aß-40 and Aß-42 be detected in patients with acute myocardial ischemia?  

  4. What could be the explanation that there is no correlation between elevated Aß-40 and Aß-42 and elevated troponin level? 

  5. Authors suggested that this biomarker may be useful in stratification in patients with NSTEMI. What is the prognostic impact of elevated Aß-40 and Aß-42 (e.g. impact on mortality or on the occurrence of non-fatal ischemic events)?

  6. What is the clinical significance of these results? 

Thank you

Author Response

Comment 1: In this cross sectional study authors analyzed correlation of plasma Aß-40 and Aß-42 with markers of myocardial injury and inflammation in patients with STEMI and NSTEMI.

Technically the manuscript is well written- methodology is good, although there is a small number of analyzed patients. Results are clearly presented. Conclusions are in concordance with findings.

My comments are:

Authors analyzed correlation between Aß-40 and Aß-42 and cholesterol, HDL, LDL. These are not biomarkers of myocardial injury nor inflammatory biomarkers, so in my opinion correlation with these biomarkers is not of clinical importance.

Response: Although, indeed, cholesterol, HDL, and LDL are not biomarkers of myocardial injury nor inflammatory biomarkers, the circulating levels of these metabolic risk factors are part of the evaluation of long-term risk before discharge in patients with acute coronary syndrome according to recent 2023 guidelines in the management of acute coronary syndrome (Supplementary data: 8.2.2. Management of acute coronary syndrome during hospitalization). Evidence shows that cholesterol affects amyloid peptide processing, but circulating levels of lipids may vary in the first days post-MI. Keeping in mind this observation, we cannot exclude the participation of these lipids in the processing of beta amyloid. Due to this, we considered it interesting to look for whether there was a relationship between the levels of these lipids and the isoforms Aβ-40 and Aβ-42 in our patients. Although our results showed no correlation, we believe lipid profile should be considered in the analysis, given that cholesterol, oxLDL, HDL, and different lipoproteins could have local effects on the cardiovascular bed and the development of atheromatous plaque..

Comment 2: Table 2. NYHA class- in patients with acute coronary syndrome we usually classify heart failure according to Killip-Kimball class only.

Response: Upon admission to the coronary unit, patients are stratified using several scales. We agree that the Killip-Kimball stratification allows the evaluation of cardiac function, similar to the NYHA scale, but allows for establishing a prognosis of the evolution of the condition and the probabilities of death in the first 30 days after myocardial infarction. Because of that, we consider that there is no problem in removing the NYHA values from the results table since it does not provide relevant information. In any case, we did not observe significant differences in either of the two stratification scales between the STEMI and NSTEMI groups.

Comment 3: Authors stated that elevation of  Aß-40 and Aß-42 in serum precede the liberation of myocardial injury enzymes. Since hs-troponin assay can detect elevated troponin level one hour after the symptom onset. When can elevated levels of Aß-40 and Aß-42 be detected in patients with acute myocardial ischemia?  

Response: Previous experimental studies have demonstrated that a peak of soluble amyloid precursor peptide(sAPP) level preceded the liberation of myocardial injury enzymes (Kitazume S. et. al. 2012, 2013). A pro-inflammatory state may increase circulating sAPP, which may favor the production of main isoforms of amyloid beta peptide (Aβ-40, Aβ-42). Aβ peptides can activate the injury endothelium response, enhancing the formation of atheromatous plaque development.

A study explored Aβ peptide levels in patients without clinically overt cardiovascular disease and reported an association with cardiac stress and injury biomarkers (Stamatelopoulos K. et. al., 2018). Considering this data, elevated Aβ peptide may be determined before, during, and after acute myocardial ischemia. Further longitudinal studies are required to explore the kinetics of Aβ in the natural evolution of the disease.

Comment 4: What could be the explanation that there is no correlation between elevated Aß-40 and Aß-42 and elevated troponin level? 

Response:The cut-off values ​​for troponin I and T concentration are described very well. High sensitivity troponin-I can be detected in serum early in the first hours of myocardial injury, achieving maximum peak level between 12 and 48 hours after symptom onset. Amyloid peptides cut-off values ​​have been described in the context of Alzheimer's disease, but cut-off values ​​applicable to acute coronary syndrome have not been established at the time of our study. Since the concentration of the amyloid peptide can increase even before the onset of symptoms of acute coronary syndrome, the precise moment at which the maximum peak in the concentration of AB40 and AB42 peptides occurs is unknown. Furthermore, no difference in hsTnI level was observed between the STEMI and NSTEMI groups, agreed with our study. The sample size may influence the relationship between the concentration of AB40 and AB42 and troponin.

Comment 5: Authors suggested that this biomarker may be useful in stratification in patients with NSTEMI. What is the prognostic impact of elevated Aß-40 and Aß-42 (e.g. impact on mortality or on the occurrence of non-fatal ischemic events)?

Response: NSTEMI patients are a vulnerable population among those with acute coronary syndrome. Adequate risk stratification is necessary to make a clinical decision, regarding better treatment or whether invasive therapy is required. Criteria and scales can facilitate the location of high-risk individuals for therapeutic decision-making. The GRACE scale has been a useful tool for locating this patient population and the association with a blood biomarker can improve this scale. According to the studies of Stamatelopoulos K et al.(2018), AB40 peptide integrated into the GRACE scale can better reclassify high-risk patients. We observed that NSTEMI patients have higher AB42 values, so the evaluation of AB42 level could have application when integrated into the GRACE scale.

Comment 6: What is the clinical significance of these results? 

Response: We know that natriuretic peptide can present changes in its levels due to the demand of the myocardium. When the insult exists, NP levels rise suddenly due to increased left ventricular stretch and filling pressure. NT-pro BNP, the stable precursor of the peptide, adds prognostic information in conjunction with cardiac troponins on the risk of mortality and acute heart failure. Considering that Aβ40 and Aβ42 can be elevated before the clinical manifestations of ACS and they have a positive correlation with NT-proBNP in acute coronary syndrome, the measurement of AB40 and AB42 peptides could place patients at high risk of mortality, together with other biomarkers such as NT-proBNP and cTn.

Round 2

Reviewer 1 Report

Comments and Suggestions for Authors

It can be seen that the changes requested from the authors were made carefully. In this form, it can be seen that the article is better, the existing gaps have been closed and the article is more elite.